# New Opportunities for Organic Semiconducting Polymers in Biomedical Applications

**DOI:** 10.3390/polym14142960

**Published:** 2022-07-21

**Authors:** Kyunghun Kim, Hocheon Yoo, Eun Kwang Lee

**Affiliations:** 1Samsung Advanced Institute of Technology (SAIT), Suwon 16678, Korea; kkhspecial@gmail.com; 2Department of Electronic Engineering, Gachon University, 1342 Seongnam-daero, Seongnam 13120, Korea; 3Department of Chemical Engineering, Pukyong National University, Busan 48513, Korea

**Keywords:** organic semiconductors, biomedical engineering, chemical sensors, biosensors, field-effect transistors

## Abstract

The life expectancy of humans has been significantly elevated due to advancements in medical knowledge and skills over the past few decades. Although a lot of knowledge and skills are disseminated to the general public, electronic devices that quantitatively diagnose one’s own body condition still require specialized semiconductor devices which are huge and not portable. In this regard, semiconductor materials that are lightweight and have low power consumption and high performance should be developed with low cost for mass production. Organic semiconductors are one of the promising materials in biomedical applications due to their functionalities, solution-processability and excellent mechanical properties in terms of flexibility. In this review, we discuss organic semiconductor materials that are widely utilized in biomedical devices. Some advantageous and unique properties of organic semiconductors compared to inorganic semiconductors are reviewed. By critically assessing the fabrication process and device structures in organic-based biomedical devices, the potential merits and future aspects of the organic biomedical devices are pinpointed compared to inorganic devices.

## 1. Introduction

Organic semiconductors are materials that can replace inorganic silicon semiconductors owing to their flexibility and lightness, tunable optoelectronic properties [1] through sophisticated molecular structure control, and low-cost processes. On the other hand, the commercialization of organic transistors and organic biosensors as next-generation electronic devices has been impeded by their lower charge carrier mobility than that of inorganic transistors and their intrinsically operational instability. This low charge carrier mobility has been attributed to poor transfer integral and significant energy loss when charge carriers transport through molecules [2]. Recently, research on the development and the improvement of high-performance organic electronics based on organic molecules and polymer semiconductors has been very active, and considerable advances in their electrical properties have been made. Scientific reports on improving the charge carrier mobility of organic semiconductors have revealed a variety of fundamental and technical approaches that exceeded the mobility of 10 cm^2^·V^−1^·s^−1^, or even higher than that of amorphous silicon [3]. For example, high-performance organic/polymer semiconductors are realized by modifying their π-conjugation length and degree at the molecular level. In addition, the energy loss in organic/polymer semiconductors is reduced further by inducing a high degree of molecular packing and minimizing their grain boundaries in fabrication steps. Thus, organic semiconductors are a key material for implementing next-generation flexible/wearable displays, smart cards, and chemical and biosensors.

One of the key advantages of using organic semiconductors in biomedical engineering applications is their better chemical compatibility in various synthesis steps compared to inorganic material-based semiconductors [4,5,6]. Therefore, electrical properties can be modified through synthetic routes depending on the selection of organic solvents, biodegradability with a proper molecular design, and the selection of functional groups in a molecular structure with a large degree of freedom (Figure 1).

Low-cost manufacturing techniques are an ongoing and ever-present research topic in the modern electronics industry. Conventional Si-based electronic devices require high-tech resources and time-consuming fabrication processes for high performance. Normally, high temperatures, around over 500 °C, are used in the film formation process (such as physical/chemical vapor deposition) to form inorganic electronic component layers on the top of a heterogeneous component. In contrast, the film formation conditions of organic semiconductors are less harsh than those used for inorganic semiconductors. A low temperature (under 200 °C) and simple fabrication processes are utilized widely for organic thin film formation, e.g., drop casting, spin coating, and dip coating. A low temperature allows the fabrication of flexible/wearable devices using polymeric substrates prone to damage by high temperature. Thus, the low-cost mass production of flexible/wearable devices for biomedical applications could be realized.

In contrast to inorganic semiconductors, organic/polymer semiconductors are intrinsically flexible or stretchable, depending on their chemical structure. The Young’s modulus of organic/polymer semiconductors ranges from 0.1 to 1 GPa [7]. On the other hand, the Young’s modulus of inorganic semiconductors ranges from several tens to hundreds of GPa, depending on their mechanical structure. Organic/polymer semiconductors are promising materials in biomedical applications in which flexibility and stretchability are prerequisites. In terms of the mechanical properties, organic/polymer semiconductors are much lighter than inorganic ones. The lightness allows devices based on the organic/polymer semiconductors to become mechanically imperceptible when fabricated into thin-film-based wearable devices [8].

One of the important advantages of organic/polymer semiconductors is their chem/biosensing characteristics in electrochemical applications to detect biological analytes from a human body. A variety of electrochemical sensing is performed in aqueous systems, in which reduction and oxidation (redox) reactions generate electrical signals. Electrical signal characteristics are influenced significantly by morphological structure, the molecular design of organic semiconductors, and the interaction between the semiconductor and the biomolecular analytes [9]. The main electrical signal detection mechanisms can be categorized into direct and indirect detections. An electrical signal can be generated by either a direct or indirect interaction between an organic semiconductor and a biomolecular analyte. The former can measure a clear electrical signal well because of direct sensing and low cost (no further functionalization is needed on the active channel). The latter has the advantage of the chemical modifiability of the organic semiconductor and long-term use, because it can avoid direct contact between the organic semiconductor and the biomolecular analyte.

Conventional Si-based biomedical devices show poor biocompatibility owing to their toxicity and short reliability in an in vivo system [10]. The most distinguishable merit of organic/polymer semiconductor devices in biomedical applications is their implantation in a human body without rejection. For example, functionalizing organic semiconductors with amine groups improves biocompatibility because of cationic charging around the amine group [10,11,12,13]. In addition, the biocompatibility of a variety of organic/polymer semiconductors and conductors (such as poly(3-hexylthiophene-2,5-diyl) (P3HT), 6,13-Bis(triisopropylsilylethynyl)pentacene (TIPS-pentacene), 3,6-bis(5-(benzofuran-2-yl)thiophen-2-yl)-2,5-bis(2-ethylhexyl)pyrrolo [3,4-c]pyrrole-1,4-dione (DPP(TBFu)_2_), and poly(3,4-ethylenedioxythiophene) polystyrene sulfonate (PEDOT:PSS)) was evaluated in terms of stability and cell adhesion [14]. All the materials were electrically stable in an aqueous medium with sufficient protection. The cell adhesion ability for each organic semiconductor depends on chemical structure and composition. Although the cell adhesion of organic semiconductors is intrinsically not as good as that of standard culture plastics owing to their hydrophobicity, the cell adhesion property can be improved by surface functionalization with a self-assembled deposition of collagen IV.

Although organic/polymer semiconductors are still far from commercialization for high-end electronic devices because of their poorer electrical performance than their inorganic counterparts, research on organic/polymer semiconductors is growing because of their tailorable charge transport characteristics according to their surrounding environment, which means that they have potential for chem/bio sensors and logic devices. Many different types of bioanalytes can be detected by changing the major charge carrier polarity (hole or electron) of organic semiconductors. Various external stimulations, such as light, heat, pressure, and chemicals, plus the fine control of the interfacial conditions, can change the polarity of organic semiconductors. For example, self-assembled monolayers (SAMs) can be used as an interfacial layer between the organic semiconductor and metal electrodes [15]. A thiol-terminated SAM can functionalize an Ag electrode in a solution-processable way. The functionalized Ag electrode alters the injection property of the organic semiconductor, which results in the polarity change.

The practical utilization of organic semiconductors is still in its infancy. Especially, the detection of biosignals using an organic semiconductor as an active component is not comparable to the use of inorganic counterparts [4,16]. However, the electrical performance of organic semiconductors can be enhanced by applying the three-terminal electrode system, e.g., transistors. This review outlines the recent development of various organic polymer semiconductors that are mainly used in field-effect transistors in biomedical applications compared to previous papers. Section 2 presents the organic semiconductors used in biomedical applications using *p*- and *n*-type charge transport. In addition, ways to enhance the electrical performance of organic semiconductors are also presented. Some fabrication techniques for organic semiconductors are introduced in terms of low-cost solution processes and large-area fabrication in Section 3. Section 4 reports a variety of device structures of organic field-effect transistors (OFETs) and organic semiconductor-based chemical sensors. Organic semiconductors can transduce several external stimuli into electrical signals. In Section 5, various device platforms, including OFETs on unconventional substrates and OFET-based biomedical sensors, will be presented. Finally, the critical issues of organic semiconductors in biomedical engineering applications will be discussed in the Conclusion.

## 2. Materials: Organic Semiconductors in Biomedical Engineering

Research on organic semiconductors in biomedical engineering has focused on synthesizing high-performance organic semiconductor materials. Recently, improved device performance in biomedical engineering has been reported via various thin film formation technologies, the development of an insulating layer, the optimization of the device structure, and improvement of the synthetic material routes.

The electrical properties of organic semiconductors can vary significantly depending on their molecular structures and molecular arrangement in the active channel. In particular, the polarity (e.g., *p*-type, *n*-type, and ambipolar) of organic semiconductors can be modulated by how they are stacked at the molecular level. In addition, using one-dimensional (1D) organic semiconductor structures formed by supramolecular self-assembly (SA) can enhance electrical performance by increasing the molecular crystallinity in which a misaligned molecular packing is minimized within the structure.

The research trends and prospects on various organic semiconductor materials used widely in biomedical electronic devices are discussed in this section. This section is mainly divided into the introduction of *p*-type organic semiconductors and *n*-type organic semiconductors. Their fundamental chemical structure design according to their polarity and functional groups constituting the molecule will be discussed at the molecular level. Owing to the maturity of organic semiconductors in FETs and biosensors, this paper introduces the technological trends of organic semiconductors in high-performance OFETs and biosensors.

### 2.1. p-Type Organic Semiconductors in Biomedical Engineering

Organic semiconductor materials are classified into *p*-type and *n*-type semiconductors according to the type of charge carriers (polarity) that contribute to the flow of current, and unipolar and ambipolar semiconductors are classified according to the combination of carriers. In addition, organic semiconductor materials are classified into molecular (Figure 2) and polymeric (Figure 3) semiconductors according to the repeated units contained.

In *p*-type semiconductors, a hole is used as a major charge carrier. The hole injected from the source electrode moves through the molecular orbital with the highest occupied molecular orbital (HOMO) energy level, allowing the current to flow into the electronic device. In general, the HOMO level, ranging from –4.5 to –5.5 eV, of organic semiconductor materials, is similar to the work function (WF) value of many commercially available metals (e.g., Ag, Au, Cu, Ti, Cr, Ni, and Pt), so hole injection from these metal electrodes is reliable. Therefore, stable charge transfer is possible compared to an electron in air. For this reason, in the development of organic semiconductor materials thus far, *p*-type semiconductors have outperformed *n*-type semiconductors, and relatively higher hole mobility has been reported compared to that of its counterparts. Representative *p*-type organic semiconductors include pentacene, fused aromatic compounds, oligothiophene, and rubrene. The thin-film formation of *p*-type organic semiconductors or the fabrication of their single crystals through vacuum deposition, solution processing, or physical vapor deposition have been used to employ these *p*-type organic semiconductors in devices as an active layer.

Research into designing the molecular structure, the morphology control of semiconductor thin films, modification at the dielectric–semiconductor interface, and the formation of microstructures and process optimization improves the electrical performance of *p*-type organic semiconductor-based transistors and biosensors. In particular, in the 2000s, high-performance organic semiconductors surpassing the charge mobility of amorphous silicon (1 cm^2^·V^−1^·s^−1^) were reported. For example, pentacene, a representative example of a *p*-type organic semiconductor, exhibits a hole mobility ranging from 1 to 10 cm^2^·V^−1^·s^−1^ depending on the deposition and dielectric–semiconductor conditions [17].

A research team at Pohang University of Science and Technology introduced a small molecule called m-bis(triphenylsilyl)benzene (TSB3) as a molecular insulator to improve the interfacial properties between the insulating layer and pentacene in semiconductors [18]. A high hole mobility of 6.3 cm^2^·V^−1^·s^−1^ was realized in the TSB3/pentacene heterojunction structure in the active channel. A low glass transition temperature (T_g_, 33 °C) of TSB3 induces a soft rubbery phase at the interface between TSB3 and pentacene. The soft rubbery phase helps reduce the number of crystal grains, improving the electrical performance. In particular, the surface phase separation formed a spontaneous nanoporous pentacene thin film, which can be applied as a high-performance chemical sensor. In a similar study, a record hole mobility of 15–40 cm^2^·V^−1^·s^−1^ was reported in a pentacene single-crystal thin film grown through physical vapor deposition (PVD) by introducing 6,13-pentacenequinone with a molecular structure similar to pentacene as an interface control material [19].

Fused aromatic compounds of organic semiconductors offer strengthened air stability by improving the degree of crystal packing and molecular orientation. Typically, a thienothiophene-based compound fused with benzene and naphthalene shows excellent electrical performance because of the improved crystal packing [20]. In addition, thin films can be manufactured via a solution process because they have an alkyl chain substituent as a solubilizing group at the benzene end group. Unlike pentacene, the lateral CH-π intermolecular interaction is disturbed by the influence of the alkyl chain in the molecules, resulting in a two-dimensional π-stack structure. Therefore, thus, more improved mobility can be expected than that of pentacene derivatives.

A solution-processable *p*-type organic semiconductor, 2,7-Dioctyl [1]benzothieno [3,2-b][1]benzothiophene (C_8_-BTBT), can be fabricated into a single-crystal thin film by inkjet printing. The maximum hole mobility from a single crystal is 31.3 cm^2^·V^−1^·s^−1^ [21]. Professor J. Huang and Z. Bao’s research team reported very high hole mobility of 43 cm^2^·V^−1^·s^−1^ in C_8_-BTBT thin films using a novel off-center spin-coating method [22]. They induced molecular orientation using centrifugal force generated when the organic semiconductor solution was dropped outside the spin-coater chuck on a rotating substrate. Furthermore, improved electrical performance was obtained by reducing the electrical trapping sites at their interface by introducing a vertical phase separation of a solution blended with C_8_-BTBT and polystyrene (PS). In a similar study, a research team led by Professor K. Takimiya and J. Takeya in Japan developed an air-stable *p*-type organic semiconductor derivative of dinaphtho [2,3-b:2′,3′-f]thieno [3,2-b]thiophene (DNTT). They fabricated a patterned crystalline thin film with a hole mobility of 11 cm^2^·V^−1^·s^−1^, in which the gradual movement of the liquid–air boundary when the solvent evaporated could simultaneously control the direction and location of crystal growth. The hole mobility was improved by a factor of four compared to the spin-coated one. A rubrene single crystal was used as a *p*-type organic semiconductor to understand the anisotropic property of charge transfer according to the lattice orientation of the crystal. The dielectric contact-free intrinsic mobility reached 40 cm^2^·V^−1^·s^−1^ [23]. In one biosensing application, an electrolyte gate FET was fabricated using C_8_-BTBT/polystyrene as a semiconductor and a blend of BTBT-biotin and a siloxane dimer of BTBT as an influenza virus-detecting layer, which was modified with streptavidin and the DNA aptamer RHA0385 [24].

Although polymer semiconductors have relatively low crystallinity compared to organic molecules, large-area printing is possible through solution processing, such as inkjet printing. Their excellent mechanical properties make them highly attractive as next-generation electronic materials. Poly(3-hexylthiophene) (P3HT), a representative *p*-type polymer semiconductor, is used widely in transistors and various applications, such as solar cells and sensors, because of its excellent optical and electrical properties [5,6]. In general, P3HT shows a low hole mobility of 0.001–0.1 cm^2^·V^−1^·s^−1^. Many studies have examined the molecular weight, regioregularity, solvent, thin-film morphology, thickness, manufacturing processes, humidity control, and design of side chain groups to improve the fundamental charge mobility. As a result, many new high-performance conjugated structures in polymer semiconductors have been developed.

Organic dye-based diketopyrrolopyrrole (DPP) and thienoisoindigo (TIIG) structures are based on a conjugated bicyclic lactam structure. Hence, their planar structures facilitate backbone alignment and induce strong intermolecular π-bonding. They were mainly introduced as backbone units of polymer semiconductors. Most polymer semiconductors are developed through donor(D)–acceptor(A)-type molecular design; electron-deficient DPP and IIG structures have been used as effective acceptors. A donor–acceptor polymer in the alternating structure can form a partial charge transfer state in the ground state and induce a very small π–interplanar distance. Furthermore, a thienyl-DPP structure in which thiophenes are linked covalently on both sides of the DPP structure has been designed, further inducing the flattening of the main chain in the molecule and improving intermolecular packing through the intra-interaction between oxygen in the carbonyl group and sulfur in the thiophene.

On the other hand, a commercially available conducting polymer, PEDOT:PSS (Figure 4), has been utilized in organic electrochemical transistors for the application of biomedical engineering.

The PEDOT:PSS is solution-processable with water and is a blended system of PEDOT as a conducting polymer and PSS as a dopant. The PEDOT:PSS, originally, is in an oxidized state with high conductivity, but its conductivity can be modulated (decreased) by a gating effect in the electrochemical transistor. Below is a typical reaction equation of PEDOT:PSS in an electrochemical transistor depending on the gate voltage condition:PEDOT:PSS + M^+^ + e^−^ ⇌ PEDOT + M:PSS
where M is a cation and e^−^ is an electron in the electrolyte of the electrochemical transistor. Under the zero-gate voltage, a high concentration of M^+^ in the electrolyte is presented, which results in high conductivity in PEDOT:PSS. On the other hand, applying (increasing) the gate voltage, PSSs are oxidized with Ms, and PEDOT:PSS becomes less conducting compared to the one without a gate voltage applied, as shown in Figure 4b. During the operation of the PEDOT:PSS-based organic electrochemical transistor, a steady state of redox reaction occurs throughout the interfaces of the PEDOT–electrolyte and the source–PEDOT–drain region. In particular, both electronic species (electrons and holes) and ionic species are transported to the region of the PEDOT–electrolyte. On the other hand, only electronic species are transported to the region of the source–PEDOT–drain. Achieving a high on/off ratio in a PEDOT:PSS-based organic electrochemical transistor is troublesome. Intrinsically, PEDOT:PSS is an organic conductor material that can flow an electrical current under a zero-gate bias. Thus, the on/off ratio with respect to the gate voltage is not as high as that of a typical organic semiconductor material. Increasing the width of the gate channel while maintaining the length of the gate channel would improve the on/off ratio [25]. The highest on/off ratio obtained was up to 10^3^ when a width-to-length ratio of the channel was larger than 10 times.

A recent research paper reported that crosslinked PEDOT:PSS-based organic electrochemical sensors are chemically robust in various organic solvents, such as dimethyl sulfoxide (DMSO), n-methyl-2-pyrrolidone (NMP), and acetone [26,27]. 3-Glycidyloxypropyl trimethoxysilane (GPOS) was mixed as a crosslinker in the solution of PEDOT:PSS to crosslink the PEDOT:PSS. The crosslinked PEDOT:PSS film as the active channel in an organic electrochemical transistor was used to detect methylene blue (MB), a biomarker for a certain redox activation in response to a change in the potential of the electrolyte with respect to the gate voltage. Moreover, during patterning a thin film of crosslinked PEDOT:PSS conventional photolithography techniques can be applied without damaging the film, providing a degree of freedom in device fabrication.

### 2.2. n-Type Organic Semiconductors in Biomedical Engineering

Research on *n*-type organic semiconductors for biomedical engineering applications has lagged far behind that on *p*-type organic semiconductors. An *n*-type organic semiconductor in which electrons are major charge carriers is oxidized easily by oxygen, moisture, and ozone molecules under ambient conditions, and its performance tends to deteriorate significantly, making it relatively difficult to develop. On the other hand, since the development of *n*-type organic semiconductors is essential for manufacturing *p*-*n* junction solar cells and high-performance complementary circuits for medical use, many studies have reported improved molecular designs, device performance, and the air stability of *n*-type organic semiconductors.

The electrons in an *n*-type semiconductor device are the major charge carriers that flow through the lowest unoccupied molecular orbital (LUMO) energy level of an organic semiconductor, so the energy level engineering of the LUMO level of the organic semiconductor and the proper selection of metal electrodes is important. In general, an electron-deficient group typically has low reduction potential, resulting in a low LUMO level in *n*-type organic semiconductors. In particular, perylene diimides (PDIs) and naphthalene diimides (NDIs), representative *n*-type semiconductors, have relatively high electron affinity, high electron mobility, and chemical and thermal stability. Hence, they are being applied as the most promising electron-deficient group unit thus far. In addition, air stability can be increased by lowering the LUMO level further through the functionalization of the core or end group, or by producing high-performance *n*-type semiconductors that are stable in air by enhancing molecular packing.

The first PDI-based *n*-type low-molecular organic semiconductor-based transistor was announced by Horowitz et al. [28]. Based on vacuum-deposited PDI thin films, in which the phenyl group was substituted at the *N*,*N*′ position, the electrical performance in FETs was only 10^−4^ cm^2^·V^−1^·s^−1^ in electron mobility. On the other hand, *N*,*N*′-bis(2-phenylethyl)-perylene-3,4:9,10-tetracarboxylic diimide (BPE-PTCDI), in which the center of the PDI was replaced with an ethyl group material, exhibited a highly π-conjugated structure, reaching an electron mobility up to 0.1 cm^2^·V^−1^·s^−1^ in FETs. Furthermore, the electron mobility was improved to 1.4 cm^2^·V^−1^·s^−1^ by fabricating single-crystal nanowires by recrystallization [28].

The alkyl chains are substituted at the *N*,*N*’ position of PDI to be soluble in organic solvents and enable the solution processing of low-molecular-weight PDI derivatives. In this case, an electron mobility up to 2.1 cm^2^·V^−1^·s^−1^ was reported by improving the molecular crystallinity and optimizing the thin-film morphology. In addition, the air stability of the *n*-type organic semiconductor device was significantly improved by introducing a fluorocarbon substituent at the *N*,*N*′ position or an electron-withdrawing substituent with a high electronegativity, such as –CN, –F, or –Cl at the center position [29].

Facchetti and Morpurgo et al. synthesized N,N’-bis(*n*-alkyl)-(1,7 and 1,6)-dicyanoperylene-3,4:9,10-bis(dicarboximide)s (PDIF-CN2), which achieved high electron affinity and excellent air stability by substituting an electron-withdrawing group, –CN, to an aromatic center in a PDI derivative substituted with a fluorinated alkyl chain. A record-high electron mobility of 10.8 cm^2^·V^−1^·s^−1^ was reported in the low-temperature region (230K) in a device structure using a vacuum gap as a dielectric layer after PDIF-CN2 was grown as a single crystal using the physical vapor transport method [30].

The first NDI molecule-based transistor with an electron mobility of 10^−4^ cm^2^·V^−1^·s^−1^ has been reported [31]. Subsequently, the NDI-based molecular semiconductor N,N′-bis(cyclohexyl) naphthalene-1,4,5,8-bis (dicarboximide) showed a high electron mobility of 7.5 cm^2^·V^−1^·s^−1^ under an argon atmosphere [32]. Typically, the molecular packing and electromagnetic movement characteristics in the solid phase of the organic semiconductor can be controlled by substituting the N,N′ position and the aromatic π–central region of the imide group.

Würthner et al. developed an *n*-type semiconductor that introduced an electron-withdrawing group, i.e., a fluorinated alkyl chain, in the NDI main chain that enhanced the air stability. The observation of the optical anisotropy of the fluorinated NDI demonstrated the self-aggregation of the fluorinate group in the sheared thin film, which resulted in enhanced electrical properties. An NDI-Cl_2_ single crystal was synthesized by solvent evaporation. The crystal growth direction of NDI-Cl_2_ affected the charge transfer and charge injection directions. As a result, in transistor applications, it was possible to achieve an electron mobility (8.6 cm^2^·V^−1^·s^−1^) that was more than twice that of a thin-film transistor [33].

On the other hand, the electron mobility of a typical *n*-type organic molecular semiconductor, fullerene (C_60_), was low (0.08 cm^2^·V^−1^·s^−1^). On the other hand, a single crystal of C_60_ was fabricated using a droplet-pinned crystallization (DPC) method developed by Bao et al. The maximum electron mobility of a single crystal of C_60_ using this method was 11 cm^2^·V^−1^·s^−1^ [34]. In the DPC process, a small piece of a Si substrate was placed on another silicon substrate onto which was polymer insulator was coated. A droplet of the fullerene solution evaporated slowly by controlling the annealing temperature to form a single crystal bundle around the small piece of Si substrate. The solvent was comprised of either a single or dual solvent to handle the morphology of the single crystal. In particular, in the case of the dual-solvent-based system, in which the boiling point and surface tension were different in both solvents, solvent circulation occurred inside of the solution, and the concentration distribution became uniform inside the droplet, resulting in a ribbon-like single crystal of C_60_. This single crystal exhibited an excellent electron mobility up to 11 cm^2^·V^−1^·s^−1^ in transistors. Similar to the DPC, one-dimensional self-assembly induced by a π–π interaction of the *n*-type organic molecular semiconductor is a promising methodology to enhance electrical performance.

Many reports on high-performance PDI and NDI-based *n*-type polymer semiconductors have been published in the form of transistors and biosensors with excellent air and operational stability and electron mobility. As an example of PDI-based polymers, Marder et al. and Lee et al. synthesized poly{[N,N′-dioctylperylene-3,4,9,10-bis(dicarboximide)-1,7(6)-diyl]-alt-[(2,5-bis(2) -ethyl-hexyl)-1,4-phenylene)bis(ethyn-2,1-diyl]} (PDIC8-EB) by coupling brominated PDI with diethynylbenzene. A nanowire suspension of PDIC8-EB can be prepared after complete dissolution, filtration, and recrystallization. An electron mobility of 0.1 cm^2^·V^−1^·s^−1^ was reported in a PDIC8-EB nanowire-based transistor [35].

The electrical performances of NDI- and PDI-based polymer semiconductors were compared using a typical NDI-based polymer semiconductor, poly([N,N′-bis(2-octyldodecyl)-naphthalene-1,4,5,8-bis(dicarboximide)-2,6-diyl]-alt-5,5′-(2,2′-bithiophene)) (P(NDI2OD-T2)), that shows a higher electron mobility than P(PDI2OD-T2), a PDI-based polymer semiconductor. This is due to the extended conjugated structure of NDI with a higher electron affinity than PDI, and the high stereoregularity of the main chain. In particular, Kim and Noh et al. reported the major charge carrier polarity modification of the P(NDI2OD-T2) ambipolar polymer and the enhanced electron mobility by blending with a small amount of an organic dopant, bis(cyclopentadienyl)–cobalt(II) (cobaltocene, CoCp2) [36].

In addition, an NDI-derived semiconductor copolymerized with an electron-donating group, poly{N,N″-bis(2-octyl-dodecyl)-1,4,5,8-naphthalenedicarboximide-2,6-diyl]-alt-5,5′-(2,2′-thienylenevinylene-thienylene)} (PNDI-TVT), was developed. A high charge mobility up to 1.8 cm^2^·V^−1^·s^−1^ was obtained [37]. In particular, although the PNDI-TVT exhibited ambipolar transport behavior with Au electrodes, *n*-type transport behavior with cesium carbonate-treated Au electrodes were obtained. This was attributed to the slightly increased polymer crystallinity under electrode regions, improved electron injection, and hole-blocking properties with cesium carbonate-treated metal electrodes.

Many synthetic routes for *n*-type polymer semiconductors have been reported by introducing amide or ester groups to the polymer core. For example, in the case of a benzodifurandione-based polymer, poly(p-phenylene vinylene) (BDPPV), with the electron-accepting group in the core, its high electron mobility of 1.1 cm^2^·V^−1^·s^−1^ has been reported with a top-gated transistor [38].

Electron-deficient units, including IIG and DPP, have been utilized as effective electron-accepting units. In particular, IIG can be copolymerized with electron-deficient cores to form acceptor–acceptor-type copolymers. An electron mobility of 0.22 cm^2^·V^−1^·s^−1^ was reported in a copolymer with benzothiazole [39]. A nitrile (CN) group was introduced to the *p*-type polymer semiconductor (PDPP-TVT) to synthesize a high-performance *n*-type polymer semiconductor (PDPP-CNTVT). The CN group is an electron-withdrawing group that enhances *n*-type charge transport behavior [39]. Therefore, the highest electron mobility of 7 cm^2^·V^−1^·s^−1^ was achieved by controlling thin-film thickness and the concentration of the solution of the transistor.

This insightful point suggests the importance of systematic molecular design and the optimization of thin-film formation conditions for realizing high-performance n-type organic transistors and sensors [40].

In summary, we have included a comparison table of key advantages (pros) and current challenges (cons) for the representative organic semiconductor and inorganic semiconductors that are utilized in biomedical applications (Table 1). The overall electrical performance of organic semiconductors, including conductivity and charge carrier mobility, etc., lags behind inorganic alternatives. However, their optical and synthetic properties can satisfy the demands of biomedical devices.

## 3. Fabrication

In terms of optimizing thin-film formation conditions, aligning the semiconducting molecules in a certain direction with respect to source/drain electrodes in a FET architecture is important because the molecules have intrinsically anisotropic structures and the main charge transport pathways are in the π–π stacking direction [41]. In addition, the organic semiconductor films need to be patterned to reduce leakage current and prevent crosstalk between the neighboring FETs [42,43]. Several efforts have been made to align and pattern the organic semiconducting crystal films using various solution printing techniques, such as solution shearing, slot-die coating, soft lithography, and direct writing techniques [44,45,46,47,48]. This section introduces how these techniques work to fabricate organic semiconductor arrays for possible application to flexible biomedical devices.

### 3.1. Soft Lithography

Soft lithography is a patterning method that utilizes elastomeric stamps, such as polydimethylsiloxane (PDMS) and polyurethane acrylate (PUA) molds, to replicate structures. This technique has attracted considerable attention and has been investigated widely for decades because it enables high-resolution patterns that range from nanometer to micrometer precision; it also has low cost and high-throughput fabrication. The patterning of organic semiconductor films using soft lithography is performed by contacting PDMS stamps onto the film and increasing the substrate temperature so that the crystals in contact with the stamps can be absorbed into the stamps [49,50]. When the stamps are removed from the substrate, the film regions that were not in contact with the stamps remain on the substrate, and high-resolution patterns can be obtained. On the other hand, this method has a disadvantage in that the size or orientation of the remaining semiconductor patterns cannot be controlled because the molecules are already crystallized before the patterning process. When the molecules are not oriented in the desired direction, the OFET performance is degraded. The performance variation over the OFET array increases, and the further application of the OFET array into the biomedical application is restricted.

Methods in which organic semiconductors can be patterned and oriented in the desired direction over a large area have been devised to solve this problem (Figure 5a) [51]. First, soft lithography capable of patterning and crystallization simultaneously was introduced for organic semiconductors using solvent vapor annealing. The PDMS stamps were immersed in the solvent reservoirs to allow the stamps to absorb the solvent, and then put into contact with the as-cast amorphous semiconductor film. At this time, the film in contact with the stamps was dissolved in the solvent, and then absorbed into the stamp while the remaining patterns were crystallized by the solvent vapor in the stamps. Uniformly oriented semiconductor patterns could be induced by crystallizing each linear pattern divided into micrometer-scale lengths rather than random nucleation and crystallization for an unpatterned film over a large area. The crystal size and uniformity of the crystal orientation were improved when the aspect ratio of the patterns increased (i.e., a decrease in pattern widths), as confirmed by polarized optical microscopy and grazing-incidence wide-angle X-ray scattering (Figure 5b). An OFET array comprising the patterned semiconducting crystals exhibited a higher average field-effect mobility (*µ*_FET_) compared to the unpatterned one. In addition, the coefficient of variation decreased over an entire array, indicating that the electrical performance of the array was improved, and a reliable OFET array was achieved by virtue of uniform organic semiconductor patterns (Figure 5c,d).

As an alternative to absorbing organic semiconductor molecules into elastomeric stamps, there is another soft lithography method that uses capillary force. The concept of capillary force lithography is that pattern replication is achieved by annealing the constituent polymer film beyond the glass transition temperature (*T*_g_) while placing the elastomeric stamps onto the polymer film [52]. Wetting the stamp wall with the polymer melt lowers the total free energy. Thus, the polymer melt can rise to fill up the voids between the polymer and the mold. This patterning method using capillary force can also be applied to a small molecule semiconductor/solvent system so that the organic semiconductor solution sandwiched between the substrate and the stamp rises up the mold wall. As the solvent evaporates, the semiconductor molecules self-assemble and crystallize.

For example, C_8_-BTBT, which is a well-known solution-processable small molecule semiconductor, was dissolved in 1,2,4-trichlorobenzene, and the solution was confined between the target substrate and pre-patterned PDMS stamp (Figure 5e) [53]. As the small molecule did not have *T*_g_, the authors applied capillary force lithography to the C_8_-BTBT solution by heating the sample to 50 °C, giving sufficient mobility to the C_8_-BTBT molecules during solidification within the trenches. The confined solution formed a meniscus and rose up the mold. After solvent evaporation, the width of the resulting C_8_-BTBT line patterns ranged from 5 to 20 µm (Figure 5f). In-plane X-ray diffraction and selected area electron diffraction analysis of the patterns showed that C_8_-BTBT crystals nucleated at the walls of the PDMS stamps, which induced the directional growth of the organic crystals. At the bottom gate, top-contact OFETs with C_8_-BTBT patterns and a 200 nm thick SiO_2_ dielectric layer exhibited an average *µ*_FET_ of 0.9 cm^2^·V^−1^·s^−1^ and a highest *µ*_FET_ of 2.6 cm^2^·V^−1^·s^−1^, indicating moderately high and uniform electrical performance (Figure 5g,h). Hence, the crystalline orientation could be controlled over a large area during the patterning of the organic semiconductor films.

In addition, various capillary force lithography techniques employing PUA molds have been introduced. The PUA mold provides more accurate and high-resolution patterning because of its higher hardness than that of PDMS, and it does not swell in the presence of a solvent (i.e., impermeable) [54,55,56].

### 3.2. Direct Writing Techniques

Although soft lithography has many advantages, as described above, it requires high-cost photo- or e-beam lithography to produce a master mold. This process can also leave low-molecular-weight oligomers on the surface after making contact with the elastomeric stamp with the substrate [57]. Such contamination may alter surface roughness, surface energy, or both, affecting the subsequent process. Therefore, in recent years, direct writing techniques that print organic semiconductors in a non-contact manner without complicated and high-cost patterning processes over a large area have been investigated extensively. Among the various direct writing techniques, aerosol jet printing, inkjet printing, capillary pen printing, and electrohydrodynamic jet printing have been used to print an array of organic semiconductors.

A capillary pen writing paradigm offers an efficient patterning approach for the printing of organic semiconductors because it is simple, and dispensing organic ink is unaffected by the printing environment (e.g., ambient temperature, humidity) (Figure 6a) [58]. The capillary pen is combined with a three-axis motorized position controller for the automated and programmed printing of organic semiconductor crystal arrays. Two hundred and fifty-three dots of TIPS-pentacene and poly(dimethyl-triarylamine) (PTAA) blended semiconductor crystals and five hundred and six dots of PEDOT:PSS electrodes were printed on a 2 × 2 cm^2^ flexible polymer substrate to fabricate bottom-contact OFETs (Figure 6b). The average *µ*_FET_ was 0.025 cm^2^·V^−1^·s^−1^ with a standard deviation of 0.01 cm^2^·V^−1^·s^−1^ and an overall device yield of more than 80% (Figure 6c,d). Hence, the direct writing of organic semiconducting crystals using the capillary pen is reproducible.

In aerosol jet printing, a functional ink is aerosolized and ejected to the substrate by a carrier gas, providing high resolution as narrow as 10 µm (Figure 6e) [59]. P3HT polymer semiconductor, ion gel electrolyte gate dielectric, and PEDOT:PSS electrode materials have been printed on SiO_2_ and plastic substrates by aerosol jet printing to fabricate 45 transistors. By optimizing the printing conditions and post-annealing process for each material (e.g., thickness and width), the P3HT-based transistors showed a high average *µ*_FET_ of 1.38 and 1.35 cm^2^·V^−1^·s^−1^ on SiO_2_ and plastic substrates, respectively. Furthermore, the printed transistors exhibited stability upon gate bias stress and bending stress on a plastic substrate, suggesting that aerosol jet printing can print high-quality organic semiconductors (Figure 6f,g).

The electrohydrodynamic jet (E-jet) printing technique has attracted considerable attention recently because of its high-resolution printing. E-jet printing ejects a functional ink from the nozzle by applying a strong electric field between the nozzle and the substrate [60]. When the ink is subjected to an electric field, electric charges accumulate on the meniscus. At the critical limit of the electric field, surface charge repulsion exceeds the surface tension, changing the circular meniscus to a Taylor cone. Thus, a droplet of fluid is ejected towards the substrate. The formation of a Taylor cone makes it possible to print fine-line patterns with widths narrower than the nozzle size. For example, although the nozzle diameter is on the micrometer scale, printing hundreds of nm wide organic semiconducting nanowires has been reported.

To print organic semiconductors such as poly(9-vinyl carbazole) (PVK), P3HT, and poly{[N,N0-bis(2-octyldodecyl)-naphthalene-1,4,5,8-bis(dicarboximide)-2,6-diyl]-alt-5,50-(2,20-bithiophene)} (N2200) using E-jet printing, Lee et al. set the nozzle-to-substrate distance at less than 1 cm to suppress the chaotic whipping motion of the ejected ink commonly observed when performing electrospinning. Organic semiconducting nanowires were printed and oriented along the desired directions using computerized x-y stage movement. Printing 3.96 wt.% PVK solution in styrene using an E-jet printer resulted in 289 nm wide PVK nanowires with a regular spacing of 50 µm. Printing approximately 15 nm long nanowires took only 2 min. *p*-Type P3HT and *n*-type N2200 semiconducting nanowires were printed by mixing their solutions with poly(ethylene oxide) and PVK, respectively, to increase the viscosity. P3HT and N2200 nanowires, 780 and 248 nm wide, respectively, were printed on SiO_2_ (100 nm)/Si substrates, and their bottom-gate FETs exhibited a *µ*_FET_ of 0.015 and 0.03 cm^2^·V^−1^·s^−1^, respectively, which were comparable to those of pure P3HT and N2200-based FETs. These FETs with well-aligned *p*- and *n*-type polymer semiconducting nanowires highlight the promising applications of E-jet printing in large-area electronic applications.

## 4. Device Structure

Organic semiconductor-based devices are used in various biomedical electronic systems through structural modifications. This section revisits the recent advances in the development of biomedical applications through the structural engineering of nanoscale device geometry. Rather than using the conventional film structure of organic semiconductors, various dimensional modifications to materials have been attempted. Nanotubes, nanomeshes, nanopores, and nanofillers have allowed organic semiconductors to detect biosensing signals with enhanced sensitivity. In 2020, Park et al. presented carboxylated polypyrrole nanotubes (CPNTs) and dopamine-specific aptamers for improved biosensors (Figure 7a) [61].

Through their experiments, they showed that the smaller diameter CPNTs with a diameter size of 120 nm exhibited 100 times higher sensitivity and selectivity than the counterparts in the wider CPNTs with a diameter size of 200 nm. They implemented an electrical-detection liquid-ion-gated biosensor based on exogenous dopamine-specific aptamer-release detection where cellular dopamine-specific aptamers are released by Ca^2+^ transport through the calcium ion channels. Someya et al. presented another structure engineered for biomedical organic devices, a nanomesh PEDOT:PSS structure on polyurethane nanofibers (Figure 7b,c) [62]. Electrophysiological detection, such as electrocardiogram (ECG) signals, was demonstrated using the proposed nanomesh PEDOT:PSS electrochemical transistors, enabling on-skin electrodes with local amplification to collect high-quality physiological signals [62]. In 2021, a nanofiller-based triboelectric nanogenerator for polymer electrolytes was reported as another healthcare monitoring application using nanostructured organic devices [63]. Li et al. presented polycation-modified carbon dot-assisted polyvinyl alcohol nanocomposite polymer electrolytes, providing triboelectric effects that responded to different mechanical stimuli (Figure 7d,e) [63]. An impressive demonstration in this study was the monitoring of physiological signals and the full joint range of motion in a fast, real-time, and non-invasive manner.

A porous structure has been proposed in recent years to increase the ability of biosensing remarkably. Salleo et al. presented molecularly selective membrane-based electrochemical transistors in a nanoporous structure to allow the real-time monitoring of the human stress hormone cortisol. The device detected the cortisol hormone selectively with the analysis of real body samples [64]. In 2021, Moon et al. also proposed a porous structure with a porous ion gel composed of poly(ethyl acrylate-ran-styrene-randivinylbenzene) and the ionic liquid (IL) of 1-ethyl-3 methylimidazolium bis(trifluoromethylsulfonyl) imide ((EMI)(TFSI)), formed by a crosslinking polymerization. Owing to pressure-dependent electrochemical properties in the proposed porous ion gel, the monitoring of various human motions, such as finger bending, was demonstrated (Figure 8) [65].

Rather than the above-revisited nanostructures in organic materials, efforts to perform structural engineering at a device level were also made. In 2022, Minami et al. reported oxytocin detection at pg·mL^−1^ (part per trillion level) using an extended-gate structure with OFETs [66]. An anti-oxytocin antibody-attached self-assembled monolayer (SAM) was used in their devices; modified streptavidin on SAM was applied to immobilize biotinylated anti-oxytocin antibodies via biotin-avidin interaction. A floating-gate-structured organic device has also been proposed as another biomedical application in which temperature- and pressure-sensing are available (Figure 9a) [67]. 

Depending on the variation in temperature in the range of 18.5–50 °C and pressure in the range of 10^2^–10^3^ Pa, the measured current was linearly changed in the presented floating-gate organic device (Figure 9b). In contrast to the conventional planar-type organic device structure, a vertically integrated biosensor based on organic semiconductors was reported. In 2020, Jung et al. presented a vertically integrated inverter circuit compromising a bottom *n*-type poly{[N,N′-bis(2-octyldodecyl)-naphthalene-1,4,5,8-bis-(dicarboximide)-2,6-diyl]-alt-5,5′-(2,2′-bithiophene)} (P[NDI2OD-T2])-based transistor and a top *p*-type poly(N-alkyl diketopyrrolo-pyrrole dithienylthieno-[3,2-b]thiophene (DPP-DTT)-based transistor [68]. As an interesting approach, this study used the shared gate, located between the two transistors, for both *p*-type and *n*-type transistors (Figure 9c). The shared gate was connected to an extended gate where the detection of the lactate was carried out (Figure 9d).

## 5. Applications

The techniques for synthesizing high-mobility organic semiconductors and the large-area printing of uniformly oriented organic semiconductors have matured. Various applications using organic semiconductor-based electronics, such as driving circuits of flexible/rollable display, photodetector array, radio frequency identification tags, and wearable biomedical sensors, have been demonstrated. Among them, considerable research is being conducted on wearable biomedical sensors because of the recent increase in medical spending and the interest in personal healthcare. Devices on various substrates, such as flexible/stretchable substrates, cylindrical metal wires, or fibers, have been used as wearable biomedical sensors.

### 5.1. OFETs on a Flexible Substrate

OFETs printed on a flexible substrate have been developed as pressure sensors and for sensing the pulse of the wrist. Monolithic OFETs that directly possess pressure-sensitive components can work actively as pressure sensors. This active matrix sensor array is advantageous in reducing power consumption. Lim et al. proposed a unique OFET structure that combined centro-apically self-organized organic semiconductors on top of printed hemispheric microstructures and an elastomeric PDMS top-gate dielectric (Figure 10a) [69].

A 2,8-Difluoro-5,11-bis(triethylsilylethynyl)-anthradithiophene (diF-TESADT) semiconductor and poly(methyl methacrylate) (PMMA) insulator blended solution was line-printed using a dispenser printer. During solidification, diF-TESADT molecules were segregated vertically on top of PMMA, and self-assembled to form semiconducting crystals (Figure 10b). The three-dimensional (3D) semiconductor array was prepared on a bottom polyethylene terephthalate (PET) flexible substrate directly contacted with the PDMS gate dielectric on indium-tin-oxide (ITO)-coated top PET substrates (Figure 10c). In this structure, the OFETs operated as pressure sensors. The capacitance increased due to the decrease in elastomeric PDMS gate dielectric thickness when external pressure was applied to the OFETs. As a result, the drain current of the OFETs increased (Figure 10d). The combination of 3D microstructured organic semiconductors with a flat elastomeric dielectric layer would improve the sensitivity of the sensors, as reported previously. The proposed pressure sensors exhibited a high sensitivity of 1.07 kPa^−1^, a rapid response and a short relaxation time of 18 ms, and good stability over 1000 cycles, which satisfied the conditions for the real-time monitoring of the pulse of the wrist artery. Fabrication on the flexible substrate enabled the wearing of sensors on the wrist (Figure 10e). When observing the current modulation of the sensors integrated on the wrist, a typical pulse shape was obtained with three prominent peaks, which can be used to assess the subjects’ health (Figure 10f). Therefore, the sensors introduced in this study have demonstrated the low-cost, large-area fabrication of FET-based pressure sensors.

### 5.2. OFETs on Unconventional Substrates

The intrinsic electronic and mechanical properties of organic materials enable their integration onto a cylindrical fiber-shaped substrate, a key component of the newly emerging electronic textiles (e-textile) for wearable electronics. The fabrication of OFETs on the cylindrical substrates can be obtained using: (1) the fabrication of OFETs at the intersection of two fibers using an electrolyte gate dielectric; and (2) the fabrication of OFETs on a single-fiber substrate.

Jang et al. fabricated cylindrical OFETs using an Al single metallic fiber as a cylindrical gate substrate, cross-linkable insulating poly(4-vinylphenol) (PVP), poly(vinyl cinnamate) (PVCN) polymers as gate dielectrics, vacuum-deposited pentacene as a semiconductor, and a thermally evaporated Au as a source/drain electrode [70]. Al metallic fiber was electropolished to a lower surface roughness before being used as a gate substrate. The surface roughness was smoothed gradually for up to 10 min and increased afterward. PVP and PVCN polymer gate dielectrics were coated on the Al wire surface by a dip-coating process and crosslinked by thermal annealing and UV illumination. Dip-coated dielectric films showed a surface roughness of approximately 0.3 nm, which was low enough to grow high-quality pentacene semiconductor crystals on the surface without disturbance. Pentacene crystals vacuum-sublimed on the wire showed similar grain sizes of 1 µm compared to the ones deposited on the flat substrates. The thermal evaporation of Au with a stencil mask defined the source/drain electrodes, and the resulting OFETs showed typical hysteresis-free transfer and output characteristics with average *µ*_FET_s of 0.24 (with PVP) and 0.53 cm^2^·V^−1^·s^−1^ (with PVCN). Under bending stress, the OFETs with PVCN dielectric exhibited excellent stability compared to the OFETs with the PVP dielectric. OFETs with PVCN dielectric maintained their electrical performance even at the smallest bending radius of 1.0 cm. The superior bending stability of the OFETs with the PVCN dielectric was attributed to the more rigid and stiff thermally crosslinked PVP than photo-cured PVCN, and the main chain of PVCN may be more flexible than that of PVP.

Kim et al. also reported fibriform OFETs using Au microfibers as a gate electrode (Figure 11a) [71].

The Au microfibers were treated with polydopamine to promote adhesion. An organic semiconductor/dielectric blend solution comprising 2,8-difluoro-5,11-bis(triethylsilylethynyl)anthradithiophene (diF-TES-ADT) semiconductor and PMMA dielectric was coated onto the polydopamine-coated Au fiber. Upon solvent vapor annealing, vertical phase separation between dif-TES-ADT and PMMA and the crystallization of dif-TES-ADT molecules occurred. Source/drain electrodes were defined by depositing 100 nm thick Au dots onto the metal insulator semiconductor microfiber through a transmission electron microscopy grid mask (Figure 11b). The average *µ*_FET_ and on/off ratio of 30 fibriform OFETs were 0.19 cm^2^·V^−1^·s^−1^ and ~10^4^, respectively. The device performance was maintained up to 80% of the original values when the microfiber was bent with a bending radius down to 3.0 mm. Finally, the authors showed fibriform OFET-embedded textiles by weaving the OFETs with cotton (Figure 11c). PEDOT:PSS/graphene oxide-coated conducting threads worked as source/drain electrodes, and their fibriform OFET showed a reasonable *µ*_FET_ and on/off ratio of 0.17 cm^2^·V^−1^·s^−1^ and ~10^3^, respectively (Figure 11d).

Recently, as a more advanced concept of OFETs with an unconventional substrate, a fiber OFET was introduced. Kim et al. reported a spirally wrapped carbon nanotube (CNT) microelectrode with the desired dimension using inkjet printing and an agarose-gel-assisted transfer technique to overcome the previously reported limitations in fabricating fiber OFETs (e.g., the use of high-cost vacuum-assisted technologies or difficulty in the fine control of channel length) (Figure 12a) [72]. 

The dimension of the CNT microelectrode was controlled by the inkjet printing conditions (e.g., the number of printing layers and the interval between patterns). With the printed CNT microelectrode, the authors fabricated OFETs and organic photodiode. In particular, the photodiode comprising poly([2,6′-4,8-di(5-ethylhexylthienyl)benzo [1,2-b;3,3-b]-dithiophene]{3-fluoro-2[(2-ethylhexyl)carbonyl]thieno [3,4-b]thiophenediyl})(PTB7-Th) and [6,6]-phenyl-C71-butyric-acid methyl ester (PC_71_BM) organic semiconductors was utilized in wearable biomedical devices measuring photoplethysmography (PPG) (Figure 12b). The photodiode measured the real-time PPG signals with discernible systolic and diastolic peaks when integrated into the textile PPG bandage (Figure 12c).

The research group also demonstrated fiber OFETs with the double-stranded assembly of electrode microfibers (Figure 13a) [73]. 

Two Au microfibers, 100 µm in diameter for use as source/drain electrodes, were coated with a P3HT semiconductor using a home-built die-coating system (Figure 13b). Subsequently, they were twisted so that the P3HT channel could be located between the source/drain fibers. In addition, they used a poly(vinylidene fluoride-*co*-hexafluoropropylene) (P(VDF-HFP)) and 1-ethyl-3-methylimidazolium bis(trifluoromethylsulfonyl)imide ((EMIM)(TFSI)) ion–gel gate dielectric material as a solid electrolyte in which an IL was dispersed in a polymer matrix. The resulting fiber OFETs exhibited typical transfer characteristics with low voltage operation below 1.3 V and an on/off ratio of 10^5^. After a successful demonstration, the fiber OFETs were embedded into a textile by weaving (Figure 13c). The embedded OFETs showed negligible changes in electrical properties because the device could endure bending deformation (Figure 13d). During bending more than 1000 times, the electrical properties, including the transconductance and threshold voltage of the fiber OFETs, were maintained up to 80% with a bending radius of 2 mm (Figure 13e). Finally, the application of the fiber OFETs for the real-time monitoring of human body signals was demonstrated (Figure 13f). Among various electrophysiological signals from the human body, a recording ECG was tested. The contraction and relaxation of the heart muscle usually results in ECG signals with an amplitude of a few hundred microvolts. Therefore, the gate and source electrodes of the fiber OFETs were attached to the wrist and chest of the human subject, respectively, so that the ECG signals could be delivered to the gate electrode and amplified by the fiber OFETs. When the fiber OFETs were set to the subthreshold swing regime, where the drain current changed abruptly, the typical ECG signals containing typical P-Q-R-S-T subwaves were amplified (Figure 13g). This suggests that the fiber OFET possessed sufficient resolution to record ECG signals; thus, it can be applied to e-textiles for wearable healthcare devices.

## 6. Conclusions

This review paper has given an overview of organic semiconductors in biomedical applications. Table 2 is a summary table to compare the materials, device structures, fabrication methods, and applications in previously reported organic material-based healthcare devices.

Organic semiconductors have several advantages in mechanical and electrical aspects over the inorganic counter parts. There are apparent limitations when it comes to electrical conductivity and charge carrier mobility compared to inorganic counterparts. They are molecularly bulky, which hinders the materialization of perfect crystalline structure at a molecular level, whereas inorganic semiconductors typically consist of atomic crystalline structures, in which the packing scale is smaller than that of organic semiconductors. This irresistible material’s nature should differentiate the direction between organic and inorganic research and development. The development of inorganic semiconductors might focus on the high performance and large capacity of the final device, whereas the development of organic semiconductors might focus on the mass production and multifunctional properties. Thus, inorganic and organic semiconductors could complement each other.

In addition, recent fabrication methods under the spotlight for organic semiconductors, including molecules and polymers, have been overviewed in this review. Soft lithography and direct writing techniques are promising methodologies based on solution processibility. These enable a precise and fine patterning of organic semiconductors on unconventional substrates. Especially, soft lithography utilizes flexible and bendable elastomeric stamps to transfer certain patterns of organic semiconductors. Flexible and bendable elastomeric stamps provide the direct printing of organic semiconductors on three-dimensional curvatures. The unique processability of those methodologies reduces tremendous fabrication costs compared to those based on conventional fabrication processes such as chemical and physical vapor deposition, and thermal and e-beam evaporators, which require high-vacuum systems.

Through the survey on material properties and fabrication advantages, organic semiconductors have great potential in biomedical applications in terms of flexible and wearable medical devices. Owing to the excellent mechanical properties of organic semiconductors, intrinsically flexible electronic devices can be realized. However, several practical challenges of organic semiconductors have blocked their commercialization.

First, their unsatisfactory robustness hinders their commercialization. Organic semiconductors are physically and chemically vulnerable. The hardness—the resistance of a material to allow plastic deformation—of organic semiconductors is too low to withstand the scratches formed during daily activities. Thus, organic semiconductors with intrinsically physical robustness should be developed. Several approaches are proposed, for example, making composite materials, mixing inorganic dielectric materials, and the modification of functional groups in organic semiconductors. On the other hand, applying an encapsulation layer on organic semiconductors is a good approach to protect the underlying layers, but additional processes are needed.

Second, the reliability of organic semiconductors has fallen far behind for long-term operation and fabrication uniformity. Owing to the intrinsic degradation of organic semiconductors over continuous operation, organic biomedical devices could deliver wrong information and sensing signals over a long period of time. This is a critical flaw of organic semiconductors in biomedical applications. In addition, fabrication uniformity should be achieved for the reliability of large-area and multi-array devices. Especially, organic semiconductors in biomedical applications can be utilized in active layers of multi-sensing components by applying functionalization to organic semiconductors. In this respect, an array of unit devices are inevitable.

Lastly, biodegradability and biocompatibility are important challenges that must be managed. With increasing demand for the mass production of organic biomedical devices, technologies for the natural disposal of organic biomedical devices should be developed. One of the approaches to solve this issue is natural dissolution on the human skin surface by environmentally friendly solvents such as water. In addition, the devices are normally attached to the human body to record biosignals. Imperceptibility and aesthetic points are of great significance. In order to achieve this, reducing the device thickness and improving transparency should be studied. By focusing on the uniqueness of organic semiconductors rather than their disadvantages, organic semiconductor-based biomedical devices could be expanded for more advanced applications.

## Figures and Tables

**Figure 1 polymers-14-02960-f001:**
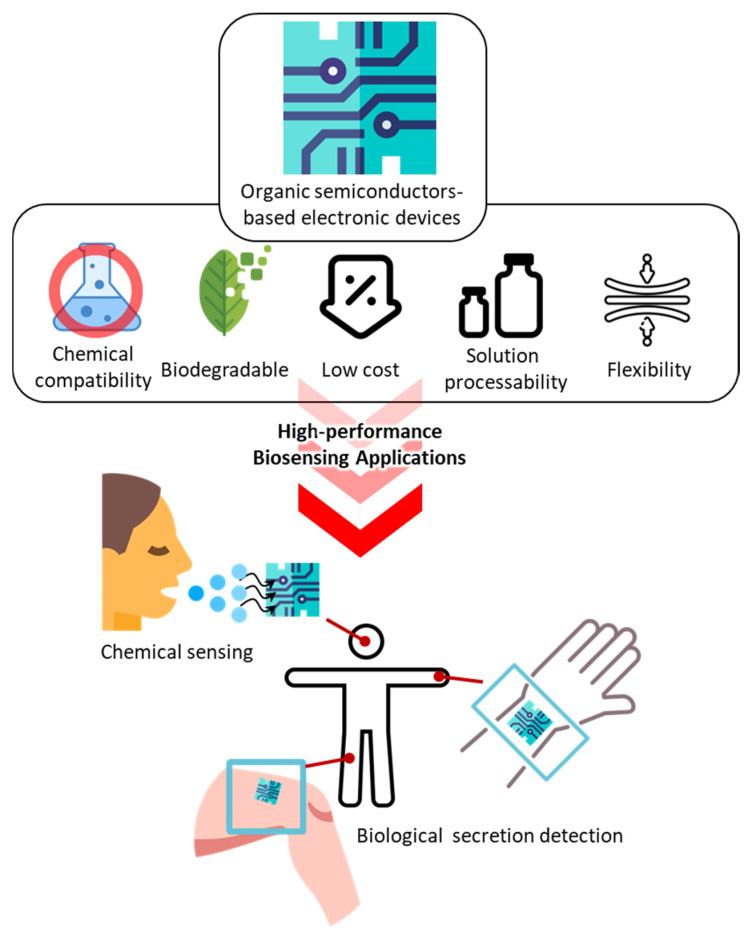
Schematic diagram of the use of organic semiconductors in biomedical applications.

**Figure 2 polymers-14-02960-f002:**
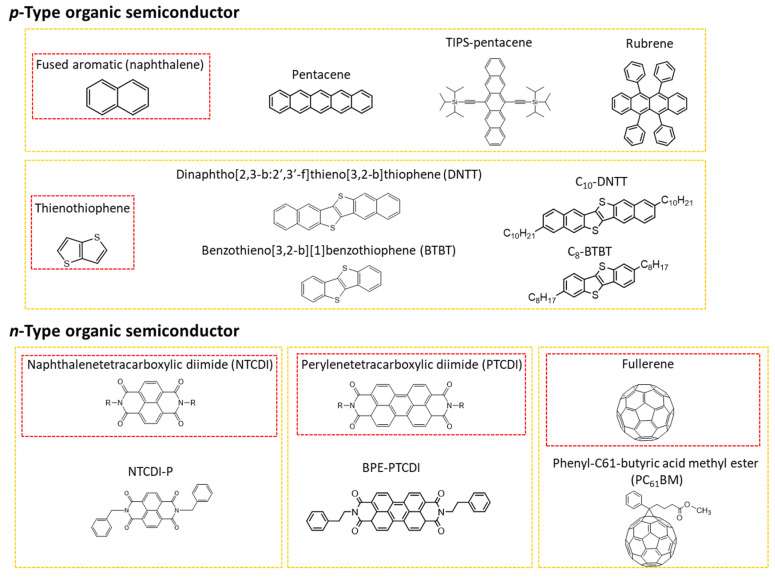
Organic semiconductors (small molecule-based) with respect to polarity (*p*- and *n*-type) in biomedical applications. A molecule in red box is a unit molecule in yellow box.

**Figure 3 polymers-14-02960-f003:**
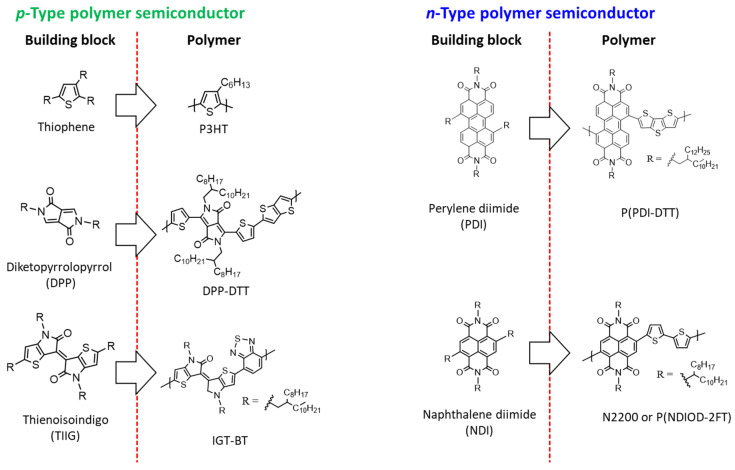
Polymer semiconductors with respect to polarity (*p*- and *n*-type) in biomedical applications.

**Figure 4 polymers-14-02960-f004:**
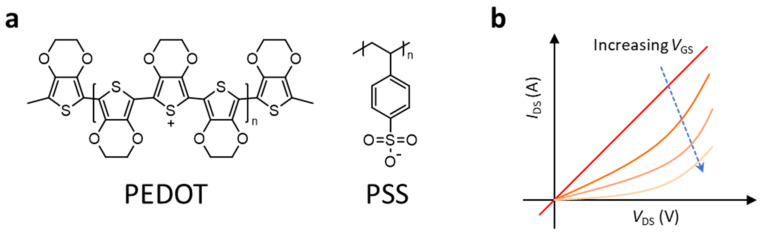
(**a**) Chemical structure of PEDOT:PSS. (**b**) The typical output curve of PEDOT:PSS depending on the gate voltage. The electrical conductance (*G* = *I V*^−1^) of PEDOT:PSS is decreased with increasing gate voltage due to a gating effect in the electrochemical transistor.

**Figure 5 polymers-14-02960-f005:**
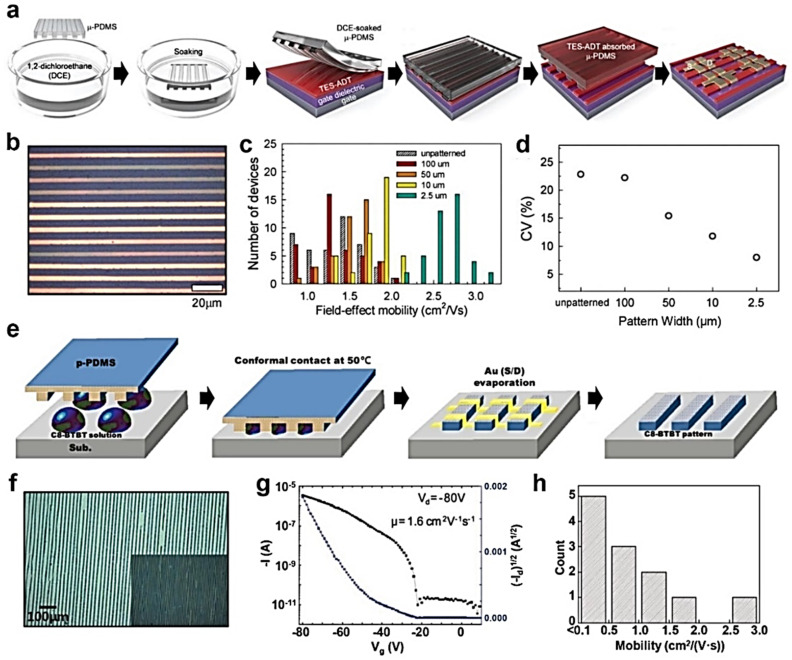
(**a**) Schematic illustration of the patterning process using PDMS stamp with 1,2-dichloroethane solvent. (**b**) An optical microscope image of the patterned TES-ADT semiconductor. (**c**) Distribution of *µ*_FET_s of TES-ADT-based OFETs and (**d**) the corresponding coefficient of variation (CV) values. (**e**) Schematic illustration of the patterning process using solvent-assisted capillary lithography with a PDMS stamp. (**f**) Optical microscope images of the patterned C_8_-BTBT semiconductor. (**g**) Transfer characteristic of the OFETs with C_8_-BTBT patterns. (**h**) Distribution of *µ*_FET_s of C_8_-BTBT-based OFETs. Reprinted from [51], Copyright 2016 with permission from Royal Society of Chemistry.

**Figure 6 polymers-14-02960-f006:**
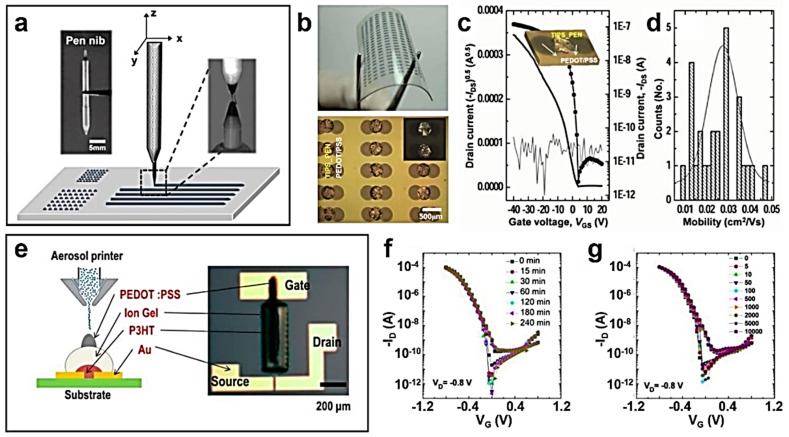
(**a**) Schematic diagram of a capillary pen writing system. (**b**) Optical microscopy images of OFETs with TIPS-pentacene dots on a flexible substrate. (**c**) Transfer characteristics of the OFETs with TIPS-pentacene dots and (**d**) distribution of *µ*_FET_s. (**e**) Schematic illustration of an aerosol jet printing system. Transfer characteristics of the printed P3HT-based OFETs according to (**f**) gate bias stress and (**g**) bending stress. Reprinted from [58], Copyright 2013 with permission from John Wiley and Sons.

**Figure 7 polymers-14-02960-f007:**
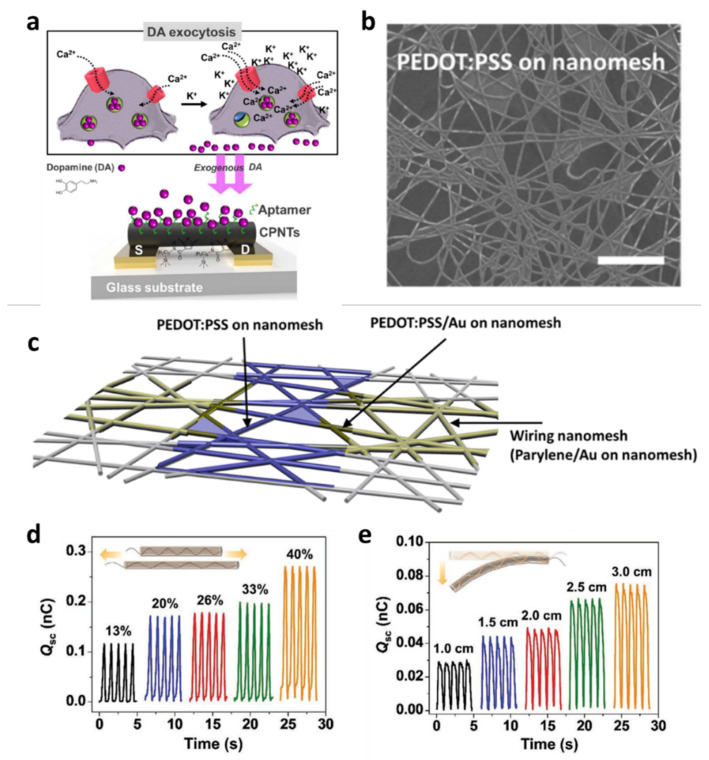
Nanotube and nanomesh structure devices. (**a**) Schematic of liquid-ion-gating for aptasensor-based oncarboxylated polypyrrole nanotubes. Reprinted from [61], copyright (2020) with permission from Springer Nature. (**b**) Enlarged image of PEDOT:PSS on nanomesh. (**c**) Nanomesh organic electrochemical transistor structure. Reprinted from [62], copyright (2020) with permission from American Chemical Society. (**d**,**e**) Measured Q_sc_ of nanocomposite polymer electrolyte triboelectric nanogenerator under different mechanical energy harvesting patterns and bending. Reprinted from [63], copyright (2021) with permission from John Wiley and Sons.

**Figure 8 polymers-14-02960-f008:**
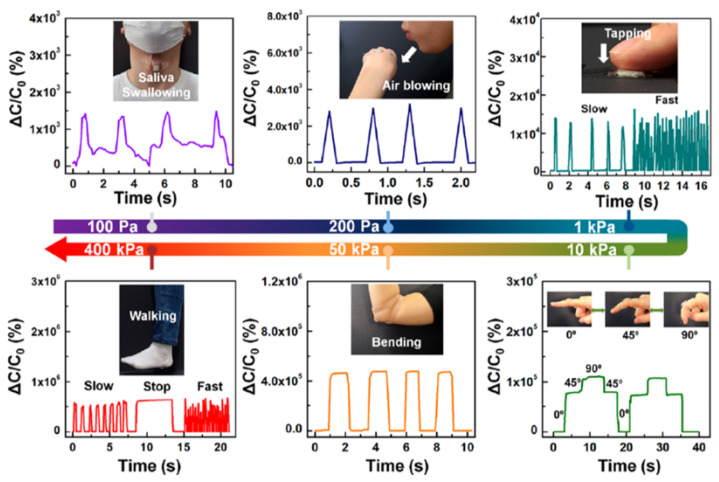
Real-time monitoring of human motions in different magnitudes of pressures based on nanopore ion gel. Reprinted from [65], copyright (2021) with permission from American Chemical Society.

**Figure 9 polymers-14-02960-f009:**
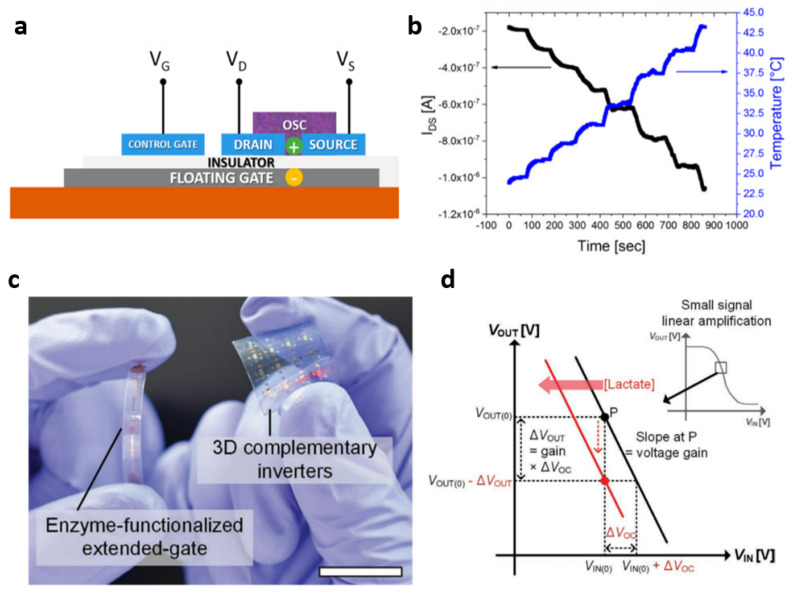
Extended gate/floating gate/3D-stacked. (**a**) Schematic of organic charge-modulated transistors. (**b**) Measured current variation recorded in real time as a function of the temperature. Reprinted from [67], copyright (2018) with permission from MDPI. (**c**) Measured image of the 3D inverter and extended-gated structured lactate detector. (**d**) Schematic of the extended-gate-type lactate detector using the complementary inverter. Reprinted from [68], Copyright (2020) with permission from John Wiley and Sons.

**Figure 10 polymers-14-02960-f010:**
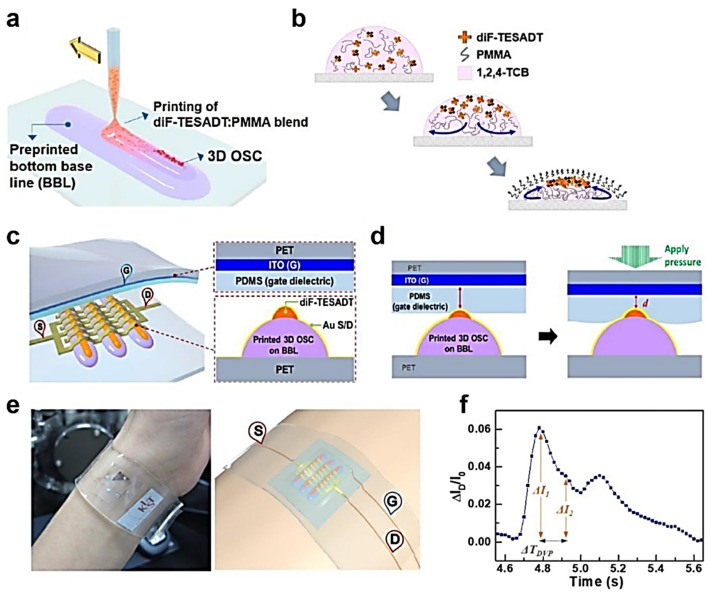
Schematic illustration of (**a**) the fabrication process of the pressure sensor, (**b**) the self-organization of the organic semiconductor, (**c**) the cross-sectional structure of the OFETs, and (**d**) the operation mechanism of OFETs as pressure sensors. (**e**) A photograph and a schematic illustration of the wrist wearing the pressure sensor using a PDMS band. (**f**) Current response to real-time pulse at constant source–drain (−60 V) and source–gate (−40 V) voltages. Three distinguishable peaks were detected. Reprinted from [69], copyright 2017 with permission from the American Chemical Society.

**Figure 11 polymers-14-02960-f011:**
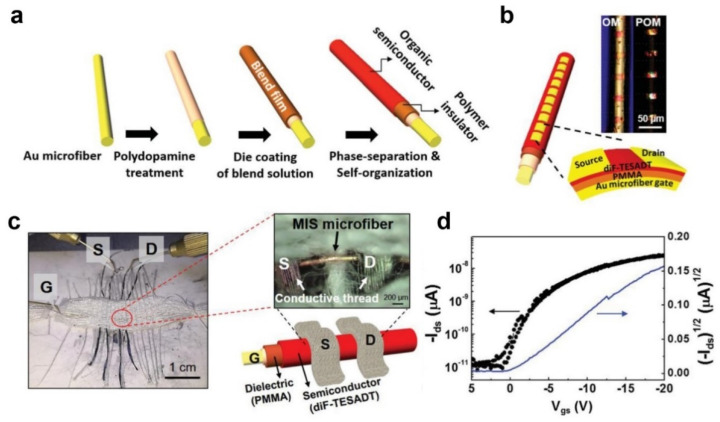
(**a**) Schematic illustration for the fabrication of fibriform OFETs using Au microfibers. (**b**) Schematic illustration and optical microscopy images of the fibriform OFETs. (**c**) Photograph, optical microscopy image, and schematic illustration of the fibriform OFETs embedded in the textiles. (**d**) Transfer characteristics of the fibriform OFETs at a source–drain voltage of −20 V. Reprinted from [71], copyright 2016 with permission from American Chemical Society.

**Figure 12 polymers-14-02960-f012:**
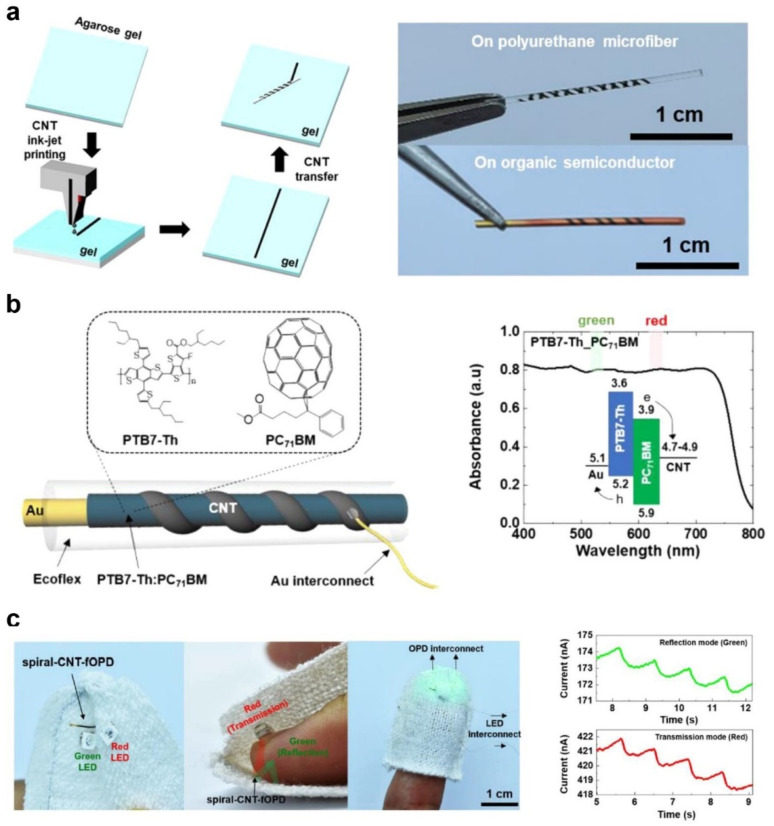
(**a**) Schematic illustration of the fabrication of CNT microelectrodes using inkjet and rolling-transfer techniques. (**b**) Schematic illustration of the device structure of an organic photodiode, chemical structure of PTB7-Th and PC_71_BM, and absorption spectrum of the PTB7-Th:PC_71_BM photoactive layer. (**c**) Photographs of green and red light-emitting diodes and photodiodes embedded in the textiles for the measurement of PPG signals and PPG signals from the photodiodes in the reflection and transmission modes. Reprinted from [72], copyright 2022 with permission from American Chemical Society.

**Figure 13 polymers-14-02960-f013:**
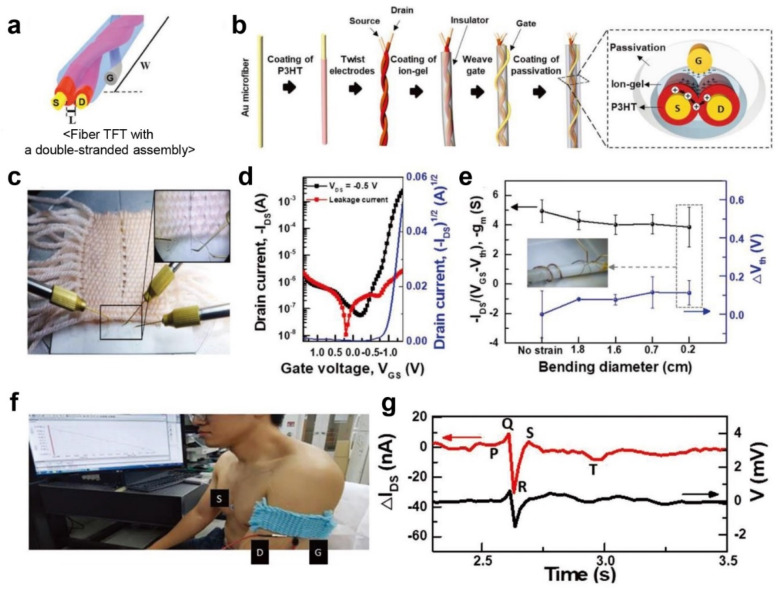
(**a**) Schematic illustration of a fiber OFET with a double-stranded assembly. (**b**) Schematic illustration of the fabrication process of the fiber OFET. (**c**) Photograph of the fiber OFETs embedded in the textile. (**d**) Transfer characteristic of the OFET. (**e**) Change in transconductance and threshold voltage according to bending diameter. (**f**) A photograph of a human subject wearing the fiber OFETs embedded in the textile for the measurement of ECG signal. (**g**) Single ECG trace from potentiometric recording (black line) and drain current (red line) from the fiber OFETs. Reprinted from [73], copyright 2019 with permission from John Wiley and Sons.

**Table 1 polymers-14-02960-t001:** A comparison table of key advantages and challenges for representative organic semiconductors and inorganic semiconductors.

Materials	Examples	Key Advantages	Challenges
Organic semiconductor	TIPS-pentacene P3HT PEDOT:PSS C_10_-DNTT C_8_-BTBT	Low cost, low temperature, large area solution process, light weight, flexibility and stretchability, tunable optical and electrical properties by synthetic routes.	Generally lower conductivity, lower field-effect mobility, lower thermal stability, lower lifetime.
Inorganic semiconductor	Si, Ge Oxide (e.g., In-Ga-Zn-O) III-V (e.g., GaAs, GaN, InN, AlN) II-VI (e.g., CdSe, CdS, ZnSe, ZnS, ZnTe)	Better conductivity, better field-effect mobility, thermal stability, long lifetime.	Hard, heavy, high-cost vacuum process.

**Table 2 polymers-14-02960-t002:** Comparison of materials, device structures, fabrication methods, and applications in previously reported organic material-based healthcare devices.

Material	Device Structure	Representative Method	Application	Refs.
**Carbon dots/** **polyvinyl alcohol**	Fiber triboelectric nanogenerator	Microwave-assisted pyrolytic reaction	Monitoring of physiological signals	[63]
**Carboxylated polypyrrole nanotubes**	Liquid-ion-gated FET	Reverse microemulsion polymerization	Dopamine detection	[61]
**PEDOT:PSS on nanomesh**	Organic electrochemical transistor	Spray coating of PEDOT:PSS	On-skin ECG signal detection	[62]
**Porous PEA-r-PS-r-PDVB**	Porous ion gel	Use of sugar template	Monitoring of human motions	[65]
**PEDOT:PSS**	Organic electrochemical transistor	Laser-patterned microcapillary	Cortisol sensing	[64]
**PEDOT:PSS/** **TIPS-Pentacene**	Floating-gate transistor	Inkjet printing	Temperature sensing	[67]
**DPP-DTT/** **P[NDI2OD-T2]**	Shared-gate structure	Spin-coating	Lactate sensing	[68]
**C6-DNT-VW**	Extended-gate transistor	Printing	Oxytocin sensing	[66]
**diF-TES-ADT**	Vertical transistor	Printing	Monitoring of the radial artery pulse	[69]

## Data Availability

Not applicable.

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
