# Peer review of "New Opportunities for Organic Semiconducting Polymers in Biomedical Applications"

_polymers, 2022, doi:10.3390/polym14142960_

Round 1
Reviewer 1 Report
The Authors summarized organic semiconductors in biomedical applications. They seems to be well familiarized with the topic, therefore the manuscript is written simply and clearly, critical discussion is the main value of this review. However, some aspects must be clarified if the manuscript is considered to be published in Polymers.
Critique points:
1. The authors should prepare Graphical Abstract to make the manuscript more attractive for potential readers.
2. The authors should revised manuscript according to Guide for authors attached in journal (e.g. Tables, figures captions, etc.). Also, all Latin phrases (via, i.e., e.g., in-situ, et al., etc…) in scientific writing should be in italics and abbreviations should be explained while using for the first time.
3. Scientific writing rules should be included https://physics.nist.gov/cuu/Units/checklist.html and Polymers internal instructions, like citation style.
4. Figure 2 and 3, all of the structures need to be named.
5. Table with summarized pros and cons of discussed organic semiconductors will be the added value for the review.
Reviewer 2 Report
The review article by Lee et al. provides brief study on the fabrication of p- and n-type organic semiconducting materials (including small molecules and polymers) for application in biomedical devices. The topic of research is interesting because semiconducting polymers are pioneering class of optoelectronic materials widely used in the area of wearable and implantable devices. Therefore, considering importance of the selected topic and systematic compilation of research, I recommend this review for publication in Polymers. My suggestions for further improvement of this review paper are as follows-
1. Considering Scope of the Journal, I suggest authors to modify the title as “New Opportunities for Organic Semiconducting Polymers in Fabrication of Biomedical Devices”. Moreover, “organic semiconducting polymers” should be focused more during discussion in the manuscript instead of “organic semiconducting materials”.
2. In the Introduction Section, authors should clearly indicate what new information is gained from this review paper in contrast to some recent reviews published in this topic (For e.g., Adv Mater, 2019, 31, 1806712, Adv Mater, 2020, 32, 2001439).
3. It is commendable that authors have compiled diverse papers and discussed them systematically. It would be better if a summarized Table showing important parameters (such as type of conjugated polymer used, specific application, fabrication method, device structure, etc.) could also be incorporated for the convenient of the readers.
4. Few structures shown in Figure 3 represent monomers and not polymers. Please correct.
5. Authors should mention the specific research group in proper manner. For e.g., “Prof. Frank Würthner et al.” should be written as “Würthner et al.” Similarly, correct “Prof. S. R. Marder et al. and Moonho Lee et al.”
6. Please obtain copyright permission for Figure 4b and mention in the caption.
7. The following important missing papers related to the topic should be cited (Adv. Mater. 2020, 32, 2001439, Adv. Mater. Technol. 2019, 4, 1900361, Microchimica Acta, 2022, 189, 83, ECS J Solid State Sci Technol, 2021, 10, 037006).
8. Whole manuscript should be check for grammatical/typographical errors.
Round 2
Reviewer 2 Report
The authors has adequately addressed all my previous concerns and thoroughly responded them by providing suitable explanation which undoubtedly improved the overall quality of manuscript. I appreciate their efforts and recommend the current version of manuscript for publication in Polymers.